# Culture in Artificial Intelligence: A Literature Review & Proposal

## Abstract

Within the last few years, there has been an explosion of various Artificial Intelligence technologies poised to change the world. However, a lot of these technologies are made by **the few** to represent **the many**. This absence of diversity amongst researchers and overall innovators creates technology that is microscopic in worldview. It brings questions of Fairness, Accountability, Transparency, and Ethics (FATE) to the forefront. Most research undertaken in the context of FATE is done within a Western cultural context, which, in turn, imparts Western values (Sambasivan et al., 2021). However, most research does not holistically address the question of the relationship between culture and AI. In this paper, we conduct a literature review of relevant research on Artificial Intelligence and culture and its importance in analyzing concepts of FATE. Additionally, we argue for and propose a definition of Culture in AI. We assume that through a combination of activation points (data collection and annotation, algorithm choice/development, problem framing, etc.), AI systems/agents perpetuate and produce culture in their dealings. This paper posits the need to situate Artificial intelligence systems within specific cultural paradigms consistent with their operative environments. We end by discussing future areas of study to be considered.

## 1 Introduction

Within the last few years, there has been rapid growth and adoption of various technologies, specifically in the realm of Artificial Intelligence (AI). However, most of these technologies are developed by **the few** to represent or to be applied to **the many**. This lack of variety among scientists, engineers, and overall innovators produces microscopic technology (both in development and application). It brings questions of Fairness, Accountability, Transparency, and Ethics (FATE) to the forefront. Most research undertaken in the context of FATE is done within a Western cultural context, which, in turn, imparts Western values (Sambasivan et al., 2021). Despite this truth, most research does not holistically address the question of culture and AI.

Many anthropologists argue that culture is a foundational aspect of all human existence. Any explanation of human behavior or extension of human behavior that ignores culture or limits it as a non-factor will almost certainly be incomplete (Evolution, 2011). Most would agree that culture can be seen and expressed through writing, linguistics, art, dance, music, etc. However, most fail to recognize that culture manifests itself fundamentally in thinking, analysis, and the scientific process despite a faulty notion of objectivity. Culture provides the framework in which thoughts and actions operate, determining the "how," "why," and "what" of an intelligent system. Artificial Intelligence is designed to model and frequently mimic natural "human" intelligence or ability. However, most ignore the subjectivity of "human" intelligence. What is deemed as intelligence can change from person to person, culture to culture. As such, we must ask the question, what is the culture of an artificially intelligent system or agent, and what positive or negative effects can it have as the system is deployed worldwide?

Additionally, the growing demand for forms of ethical machine learning (Barocas et al., 2017a; HLEG, 2019) has led researchers to propose numerous ethical principles. To increase fairness in algorithms, researchers have proposed mathematical criteria in an effort to be "precise". Yet, many of these fairness criteria have been shown to be mutually incompatible (Kleinberg et al., 2016), and these rigid formalizations are too narrow in scope (task-specific). To increase safety in these algorithms, researchers have proposed specifying safety constraints (Ray et al., 2019). However, these rules have various exceptions in the open world or require more contextualized interpretations. In an effort to make algorithms more prosocial, researchers have proposed learning/imitating traits (e.g., empathy) (Rashkin et al., 2018; Roller et al., 2020); however, these have been limited to specific traits in applications such as chat-bots (Krause et al., 2020). It is important to note that the majority of the research conducted on developing ethical and fair AI systems has been undertaken from a primarily Western cultural worldview. As noted in [8] by Birhane and Cummins, "It is possible that what is considered ethical currently and within certain domains for certain [cultures] will not be received similarly at a different time, in another domain, or for a different [culture]."

In this study, we undertake a review of the literature of significant research on Artificial Intelligence and culture, and we discuss the value of this research when it comes to understanding the ideas of FATE. In addition, we argue for the existence of Culture in AI and provide a working definition for it. In all their interactions, AI systems and agents perpetuate and construct culture through activation points (including data gathering and annotation, algorithm selection and development, problem framing, etc.). The core premise of this research is the need to embed artificial intelligence systems within particular cultural paradigms consistent with their operational contexts. We conclude by discussing potential avenues of research.

## 2 LITERATURE REVIEW

In the following sections, we discuss current research and research trends in the context of machine behavior and aligning AI and human values. We specifically chose machine behavior and value alignment as the focal points of review due to their foundational importance in the evolving landscape of AI technology. Understanding machine behavior is essential for developing AI systems that can be trusted, predicted, and harnessed effectively. Likewise, aligning AI with human values is a moral imperative to ensure that the technology is an empowering force rather than a disruptive one.

### 2.1 MACHINE BEHAVIOR

Rahwan et al. (2019) propose a question as to whether there exists an "artificial" science— a set of knowledge built about artificial objects and phenomena. This question stems from the notion of natural science- a set of knowledge built about naturally occurring phenomena. In their work, the researchers discuss the development of an interdisciplinary field of scientific inquiry focused on studying intelligent machines and systems—not just as engineering artifacts but as an emerging class of actors with distinct behavioral patterns and ecological characteristics, (Rahwan et al., 2019). This field of study is similar to the study of animal behavior by integrating both intrinsic properties (environment physiology and biochemistry) and properties shaped by the environment (ecology and evolution). Machine behavior, like animal and human behaviors, cannot, and should not, be comprehended without consideration for the contexts in which these behaviors occur. Machine behavior can only be fully understood through the integrated study of algorithms and the social context in which the algorithms operate in (Milner, 1981).

Rahwan et al. (2019) adopt and modify Tinbergen's type of question and object of study (Tinbergen, 2005; Nesse, 2013) to develop three scales of inquiry for studying machine behavior: individual machines, collectives of machines, and groups of machines embedded in a social environment with groups of humans in hybrid or heterogeneous systems.

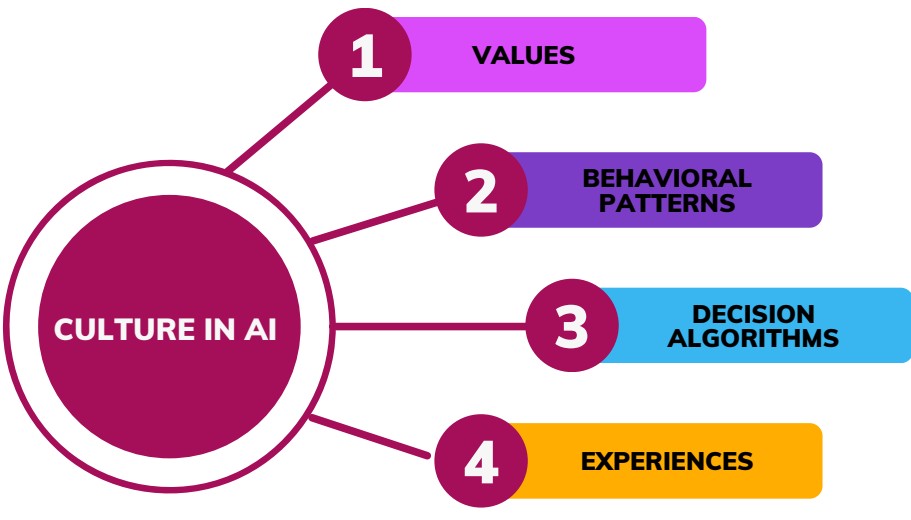

Figure 1: The four major components and activation points of Culture in AI

Nikolaas Tinbergen was awarded the 1973 Nobel Prize in Physiology for their contributions to the establishment of the field of ethology. Nikolaas Tinbergen developed four complementary dimensions of analysis that assist in explaining animal behavior (Tinbergen, 2005). These aspects provide an organizing framework for studying animal and human behavior by addressing problems regarding the function of behavior, its mechanism, its development, and its historical relationship to evolution.

While machines and animals clearly have fundamental differences, Rahwan et al. believe that the behavioral study of machines could benefit significantly from a similar categorization. Machines, specifically in the context of AI, produce behavior, undergo development that integrates information about the environment into the behavior, produce impacts that cause specific machines to become more or less common in particular environments, and represent evolutionary histories that continue to influence machine behavior through the ways in which past environments and human decisions continue to influence machine behavior.

Other authors have sought to study machine behavior by studying biased representations and predictions in AI systems. Hutchinson et al. (2020) show that representations encoded in language models can perpetuate negative and unwanted social biases from the data on which they are trained. They then show evidence of these negative biases towards persons with disabilities in two English language models: toxicity prediction and sentiment analysis. Sun et al. (2019) conduct a review of gender biases in NLP algorithms and systems as well as discuss approaches that have been taken to mitigate some of these biases. Other researchers have looked at Machine/AI behavior in settings ranging from NLP (Abzaliev, 2019; Agarwal et al., 2019; Bender, 2019; Barocas et al., 2017b; Beukeboom & Burgers, 2019) to Computer Vision (Birhane, 2022; Birhane et al., 2021; Birhane & Prabhu, 2021; Zhao et al., 2021), from healthcare (Cho, 2021; Parikh et al., 2019; Norori et al., 2021) to crime prevention (Završnik, 2020; Rejmaniak et al., 2021).

## 2.2 AI VALUE ALIGNMENT

Various authors have approached the topic of AI value alignment from different angles in the context of culture and AI. Researchers have investigated concepts ranging from aligning models with human values (Kasirzadeh & Gabriel, 2023) to Fairness, Accountability, Transparency, and Ethics (FATE) in the context of AI. In this section, we review various papers and their contribution to the topic of culture.

In their paper, Kasirzadeh & Gabriel (2023) try to answer the question of what it means to align conversational agents with human norms or values. Which norms or values should they be aligned with? This research aims to shift away from piecemeal methods for analyzing biases in agent alignment and move toward a more principal-based approach to this problem. The primary goal of the principle-based approach to conversational agent alignment is to ascertain the underlying norms and rules that direct effective linguistic dialogue. This strategy aims to characterize more clearly what ideal language communication is across a range of settings and to realize these properties in the design of conversational agents. In their work, the authors focus on three requirements that, as they posit, should be satisfied for successful human-agent conversation: syntactic, semantic, and pragmatic criteria. Additionally, they acknowledge that language performs many roles and functions in different domains. As a result, an explanation of effective communication needs to additionally consider whether or not a certain language is helpful in relation to what end, for what set of people, and in what manner. The questions raised in the above research are essential in the context of Culture and AI as they help clarify the foundation of Large Language Models and communication, a fundamental expression of culture.

Another approach to aligning AI values with human values is shown by Hendrycks et al. (2020). They developed a benchmark dataset, ETHICS, which seeks to quantify the ability of a large language model to learn the salient features of textual representations of morality in the forms of various scenarios and questions. While this is a good step in the direction of quantifying ethics, which is an outgrowth of culture, it still is limited in that it focuses on concepts of ethics and morality from a specific cultural background, namely a Western one.

Prabhakaran et al. (2022) introduce an argument for the application of human rights principles as a way to bridge the gaps in the development of ethical/responsible AI. The authors propose a multifaceted role for human rights, extending beyond a legal framework to include three additional functions. Firstly, human rights are viewed as having cross-cultural validity, capable of fostering value alignment for AI systems across diverse national and social contexts. Secondly, human rights can illuminate the shared responsibilities of various stakeholders, encompassing states, organizations, and individuals. Thirdly, human rights offer a common vocabulary and framework for technologists to address global civil society's concerns and the impacts of AI technologies. By distinguishing between human rights as moral claims, a legal regime, and a cultural movement, the paper highlights their application to responsible AI development.

From a philosophical standpoint, the authors delve into the foundational nature of human rights as intrinsic rights inherent to all individuals due to their existence on earth (regardless of ethnicity, nationality, or religion). These rights encompass safeguarding essential aspects of personhood and are grounded in shared human interests, autonomy, freedom, and the dignity of human life. Furthermore, the authors emphasize the legal dimension of human rights, which has evolved from historical atrocities, leading to the development of the Universal Declaration of Human Rights (UDHR) and subsequent international covenants.

This question of alignment is crucial as it brings to the forefront the process of deliberately developing agents within a specific paradigm of cultural values rather than a free-for-all approach that will continue to produce unwelcomed biases. However, most research on AI value alignment does not exist under a unified framework for contextualizing and defining culture.

# 3 OUR CONTRIBUTION

Our efforts in this paper are not to ascribe sentience or to unnecessarily anthropomorphize AI systems. Our focus is on elucidating aspects of AI systems that can propagate cultural values through their life-cycle.

## 3.1 DEFINITION OF CULTURE IN AI

In this work, we define Culture in AI as the following:

> **The sum total of *values*, *behavioral patterns*, *decision algorithms*, and *experiences* that characterize an AI system and are unique to that system.**

We specifically craft a broad definition of Culture in AI to avoid a narrow approach to determining what contributes to the overall operations of an AI system. The following sections discuss each key part of the definition, their importance, and more.

**Values** The values of an AI agent are represented by what the system holds as a priority. The values can be derived from initial training data, the patterns the agent is trained to recognize, and those the agent is trained to ignore. These values can be directly influenced by humans in that the impetus for the agent development, the collection of training and testing data, and the environment that the agent will operate within are determined first and foremost by the programmer/creator.

**Behavioral Patterns** The behavioral patterns are recurrent actions or decisions by a single agent or group of connected agents toward a given object or situation. These patterns serve as windows into the underlying values that drive agent behavior. By analyzing these recurrent behaviors, we can uncover the fundamental values that guide the actions of these agents, shedding light on the fundamental principles that shape the system's dynamics.

**Decision algorithms** These algorithms encapsulate an agent's systematic steps to reach conclusions or make determinations when presented with new information. They form the backbone of automated decision-making processes. This can refer to black box algorithms as well as white box algorithms. These include programmed and learned decision algorithms.

**Experiences** Agent experiences are interactions that an agent has with data from its initial training to the end of its life cycle. These interactions span a vast spectrum of encounters with information, enabling the AI agent to learn, adapt, and make informed decisions based on accumulated knowledge. Some of the major experiences that an agent may have are:

- **Initial Training**: An AI agent's experiences begin with initial exposure to training data. During this phase, the agent learns patterns, correlations, and representations present in the data, allowing it to develop an initial "understanding" of the task it's designed to perform. This stage often involves techniques like supervised, unsupervised, or reinforcement learning, depending on the nature of the task.

- **Agent Evaluation**: During this set of experiences, an agent, or model, is tested against some specific set of metrics. If the agent does not meet a qualifying threshold determined by the creators, the agent will experience model training again.

- **Agent Deployment**: Experiences during this time typically are between the agent(s) and end-users. Here, an agent should be monitored closely to determine if any problems arise as it comes into contact with unseen data. Here, lack of fairness and negative biases are typically brought to light.

- **Fine-Tuning and Personalization**: In certain applications, AI agents can be fine-tuned or personalized based on individual user interactions. For example, recommendation systems can adapt to users' preferences over time, refining their suggestions based on the user's experiences.

- **Monitoring and Maintenance**: During the experience of deployment, it may be shown that the agent needs to be modified or adjusted. During the monitoring and maintenance, a model may be retrained or adjusted in some manner.

Values, Behavioral Patterns, Decision Algorithms, and Experiences—the four components of Culture in AI—form a complex web of interactions that collectively shape how an AI system functions within a given cultural context. These components are known as "Activation Points", representing important junctures where the AI system is activated inside a particular cultural paradigm. Recognizing these activation points requires the contemplation of four major questions: ***What is the worldview of your dataset? (values) How does your algorithm/model consistently engage data: seen and unseen? (behavioral patterns) How does your model make decisions? (decision algorithms) What interactions does your model have, and how does it change or affect your overall model? (experiences).*** Understanding these activation points is critical since they are the catalysts for the AI system's behavior and the breeding grounds for deliberate and unintended biases.

**Values as Activation Points**: Values are the foundational beliefs and principles, the worldview, that guide the behavior of an AI system. These values are integrated into the system's programming and design, serving as the initial activation point that defines the system's goals and priorities (Birhane et al., 2022; Birhane, 2022; Birhane et al., 2021). As stated by Baker, "There is no "view from nowhere", no universal way to organize every object, or word, or image. Datasets are always products of a particular time, place, and set of conditions; they are socially situated artifacts. [...] If it's hard to see a dataset's values—if it feels "objective", "universal", or "neutral"—it may simply be reflecting a worldview you're accustomed to."

**Behavioral Patterns as Activation Points**: Behavioral patterns emerge from the interactions between the AI system and its environment. These patterns represent the consistent responses the system exhibits in various situations. These behavioral patterns are initially concretized during the training and testing of the model. These behavioral Patterns may change depending on the underlying architecture (e.g., human-in-the-loop, continuous learning, etc.). In this context, the AI's activation point occurs when it interprets input data and takes specific actions. For example, if an AI chatbot's behavior patterns are shaped to be friendly and helpful, its activation point involves responding to user queries with empathy and informative answers (Zhao et al., 2023).

**Decision Algorithms as Activation Points**: Decision algorithms dictate how an AI system processes data and arrives at conclusions. The activation point for these algorithms is when the AI receives new information and applies its decision-making process. Suppose a self-driving car's decision algorithm is programmed to prioritize pedestrian safety. In that case, the activation point occurs when the vehicle encounters a situation requiring it to make split-second choices to avoid accidents (Katare et al., 2022; Howard & Borenstein, 2019).

**Experiences as Activation Points**: Experiences encompass the AI system's interactions with data, training, real-world scenarios, and user feedback. These interactions refine the AI's capabilities and influence its future decisions. The activation point for experiences is when the AI engages with new data or user interactions, updating its knowledge and adapting its behavior accordingly (Getahun, 2023). For instance, a recommendation system's activation point occurs when it processes a user's preferences and refines its suggestions based on their feedback (Baeza-Yates, 2020; Saxena & Jain, 2021; Dinnissen & Bauer, 2022).

At these activation points, biases—implicit cultural attitudes and assumptions—can take root. Biases might be introduced when defining what is considered essential or relevant during the development of values. They can emerge in behavioral patterns when system responses inadvertently reflect cultural stereotypes. Biases

can also be encoded in decision algorithms if the training data contains biased information or if the algorithm represents a biased set of steps. Lastly, biases may seep into experiences if the AI system learns from biased historical data or user interactions.

It's crucial to acknowledge that biases aren't inherently negative; they can also serve as mechanisms to preserve cultural norms and values. However, harmful biases perpetuating discrimination or reinforcing unfair social hierarchies need to be recognized and rectified (Stanczak & Augenstein, 2021).

By identifying these activation points and understanding potential emerging biases, developers, researchers, and stakeholders can work collaboratively to mitigate harmful biases and promote responsible AI development. Addressing biases at these activation points contributes to creating AI systems that align with societal values, promote fairness, and respect the diverse cultural contexts in which they operate.

## 4    IMPLICATIONS OF CULTURE IN AI

We briefly discuss some implications of concepts of Culture in AI and its potential effect on the future of technological development.

The notions encapsulated in the concepts of Culture in AI bear significant implications for the trajectory of technological development, paving the way for both transformative advancements and critical ethical considerations. By examining the potential effects of these concepts, we gain insights into how the recognition and integration of Culture in AI might shape the future landscape of technology.

**Ethical Reflection and Accountability** Integrating values into AI systems prompts a reevaluation of the ethical implications of technology. As Culture in AI emphasizes fundamentally an evaluation of culture, value alignment, and ethics considerations, it encourages a proactive approach to identifying and mitigating biases, discrimination, and unintended consequences. This emphasis on ethical reflection and accountability creates a demand for transparent decision-making processes, and it incentivizes developers to anticipate potential pitfalls before they arise.

**Human-Centric Design** Culture in AI fosters a shift toward human-centric design. By understanding and focusing on cultural values and behaviors, AI systems can be tailored to serve specific user populations. This approach enhances user satisfaction and promotes inclusivity, making technology more accessible and relevant to more individuals.

**Culturally Contextualized AI** Using a Culture in AI paradigm can lead to the development of Culturally Contextualized AI. That is, AI technologies/tools that are specifically developed to operate within a specific cultural worldview rather than standing as a supposed one-size-fits-all conglomeration of cultures and experiences, leading to several unwanted and harmful biases.

**Mitigating Technological Bias** Concepts within Culture in AI contribute to the ongoing fight to mitigate negative biases. By addressing biases at the activation points of AI—values, behavioral patterns, decision algorithms, and experiences—developers can create technology that upholds fairness and avoids reinforcing societal prejudices.

Culture in AI introduces a paradigm shift in how technology is conceived, developed, and integrated into society. The implications extend beyond technical capabilities to encompass ethical considerations, societal impact, and the empowerment of individuals and communities.

## 5    FUTURE WORK

Looking ahead, the realm of Culture in AI presents a promising avenue for future research that warrants a more profound investigation into its four fundamental components. Delving deeper into these components

has the potential to unveil insights regarding the complex relationship between artificial intelligence systems and human culture. Some aspects of these components, however, necessitate a longitudinal study of AI systems and agents spanning an extended period of time. To understand the overlapping dynamics of machine behavior and culture alongside human culture and behavior, an interdisciplinary approach is essential.

In this regard, future research endeavors could be directed towards prolonged observations and analyses of AI systems and their various activation points, aiming to discern the evolution of their behavior and its impact on society. Such studies could involve computer science and AI expertise and insights from anthropology, sociology, psychology, and other relevant disciplines. By engaging multiple perspectives, researchers can shed light on how AI systems influence and are influenced by human cultural norms, beliefs, and practices. Moreover, understanding the reciprocal relationship between Culture in AI and human culture necessitates an extended temporal scope, as the interactions and adaptations of AI systems may unfold over extended periods, shaping and being shaped by the societies they exist within.

## 6 CONCLUSION

This paper conducts a literature review of concepts related to Culture in AI, such as Machine Behavior and AI Value Alignment. Additionally, we propose a definition of Culture in AI with four main components: values, behavioral patterns, decision algorithms, and experiences. We describe each part and its relevance to the overall definition of culture. We then further explore the concept of Activation Points and how each component acts as an activation point of Culture in AI. We end by discussing the implications that our definition and the growing framework of Culture in AI holds regarding the future of artificial technological development.

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
