# OpenReview forum: "Culture in Artificial Intelligence: A Literature Review & Proposal"
_ICLR.cc/2024/Conference — ICLR 2024 Conference Withdrawn Submission_

### Official Review · Reviewer_ftsk · 2023-10-25

**Soundness:** 1 poor
**Presentation:** 2 fair
**Contribution:** 2 fair
**Rating:** 1
**Confidence:** 4

**Summary:**

The authors propose a definition of Culture in AI based on a review of papers. They argue that many fairness algorithms with mathematical definitions are too rigid and narrow in scope. They draw inspiration from work by Nikolaas Tinbergenand argue that Culture in AI can be decomposed into (1) Values, (2) Behavioral patterns, (3) Decision Algorithms, and (4) Experiences.

**Strengths:**

The paper is well written and clear. The need to model and assess automated decision systems and their impact on society is also necessary.

**Weaknesses:**

No research, technical claims or research methodology is put forward. The authors argue for an informal definition of Culture in AI. They define culture, however it is unclear if the definition they are working with is original. AI is not defined, but the examples offered stem from LLMs and chat-bots. The paper is well organized, but the number of references for a literature review are too sparse. AI appears to be narrowly defined around text and chat-based systems, especially given the wealth of prior work on automated decision systems, how they shape our culture and privilege. I would have also appreciated more recent results around the cultural effects of text-to-image generative models.

The authors propose a 4-point lens by which they can study the impact of AI on culture. Such a definition can certainly be useful, however it is not directly applied to any of the works reviewed. It is therefore difficult to determine the utility of such a definition if it hasn’t been applied to even a few key examples.

**Questions:**

- Why did you focus exclusively on text and chat?
- What are competing culture and AI definitions and what their blind-spots in practice?
- Why is this the moment to discuss culture and AI and why not any other time?
- What is novel about here and now that demands such a definition?
- How is this definition connected to AI and Machine Learning? Are they the same?
- What is your working definition of AI, is it fixed or fluid? What is excluded from your definition?
- Where is Culture in AI inapplicable?
- You claim that there is a lack of variety among scientists, engineers and innovators, how do you define these groups? How do you determine homogenaity? What are the examples of group-think that you claim is anathemic to a broader cultural lens?
- How does Culture in AI intersect with regional and linguistic normative values? What are some examples which illustrate how this 4-component definition surfaces this monolithic approach to technology?

---

### Official Review · Reviewer_VFYa · 2023-11-02

**Soundness:** 1 poor
**Presentation:** 2 fair
**Contribution:** 1 poor
**Rating:** 3
**Confidence:** 3

**Summary:**

The authors discuss a few recent papers examining FATE and AI, then present a definition of culture in AI.

**Strengths:**

The paper goes over relevant papers from the AI ethics domain and tries to answer the question of what it means for an AI to have culture. The authors present some support for their framework, and present good areas for future work.

**Weaknesses:**

This paper never even bothers to define what the authors mean by AI. As it's never made clear to me what we are discussing I'm not sure how to interpret any of the arguments.

Even if I take the most vague interpretation of AI, I still find the premise difficult. We can ask about culture in sociology-techinal systems or about how tools reflect the culture they are created for/by, but saying some set of matrices have a culture is not something I can believe. Maybe for some specific set of online learning algorithms it's a colourable argument, but most AI systems don't evolve over time. The humans running them change them over time, based on the human's culture, values, etc.

I think that if the claim was less strong, and the term AI better defined then this could be a good paper. Reducing the anthropomorphize a bit more would also help, the language choice in section 3.1 I find troublingly overloaded.

**Questions:**

Why was this framed as "culture in AI" instead of "cultural context of AI"?
What exactly do you mean by AI? Is an MNIST classifier AI? Or a chess engine?
How does the presented paradigm mitigate biases that other methods of examining the data do not?

---

### Official Review · Reviewer_T5ua · 2023-11-05

**Soundness:** 4 excellent
**Presentation:** 3 good
**Contribution:** 3 good
**Rating:** 8
**Confidence:** 5

**Summary:**

This paper proposes the inclusion of values, behavioral patterns, decision algorithms and experiences as activation points to infuse Culture in AI. Authors talk about the importance of including culture into AI given how the current landscape of AI ethics is not considering any such framework and all mitigation strategies are confined to westernized world views.

**Strengths:**

Their definition of culture as it pertains to the author's efforts to make AI ethical and responsible is quite impressive. Most papers in this domain do not take pains to define what they are proposing but definitions are important, if one wants to quantify outcomes. The framework they propose is interesting and hopefully we can see it fully developed as a pilot and field tested,

**Weaknesses:**

Since this is a position paper, I can tell the authors did a thorough job of presenting their concept, its impact and need for future research. But as a theoretical paper, it was missing a possible example or examples of alignment with any current ethical issues.

**Questions:**

Please provide a description ( theoretical or otherwise) of how Culture in AI can help remove bias in any current use case. The paper was missing a hypothetical example of alignment with any of the current issues of lack of ethics, fairness, transparency or accountability in AI.
It is easy to propose these activation points as a methodology to infuse Culture into AI but without an attempt to apply ( in theory) to current issues and debate the pros and cons under the Future Work.